# Cultivation of Cowpea Challenges in West Africa for Food Security: Analysis of Factors Driving Yield Gap in Benin

**Firmin N. Anago [1,*], Emile C. Agbangba [2,3], Brice T. C. Oussou [4], Gustave D. Dagbenonbakin [4] and Lucien G. Amadji [1]**

1   Laboratory of Soil Sciences, School of Plant Production, Faculty of Agricultural Sciences, University of Abomey-Calavi, 01 BP 526 RP Cotonou, Benin; gamadji@yahoo.fr
2   Laboratoire de Biomathématiques et d'Estimations Forestières, Faculté des Sciences Agronomiques, Université d'Abomey-Calavi, 04 BP 1525 RP Cotonou, Benin; agbangbacodjoemile@gmail.com
3   Laboratory of Applied Biology Research and Study, Polytechnic School of Abomey-Calavi, Department of Environmental Engineering, University of Abomey-Calavi, 01 BP 2009 RP Cotonou, Benin
4   Laboratory of Soil Science, Water and Environment, Research agricultural Center of Agonkanmey, National Institute of Agronomic Research of Benin, 01 BP 988 RP, Cotonou, Benin; briticoss@gmail.com (B.T.C.O.); dagust63@yahoo.fr (G.D.D.)
*   Correspondence: firmin.anago@gmail.com; Tel.: +229-97750183

**Abstract:** Feeding the world in 2050 requires us to find ways to boost yields of the main local crops. Among those crops, cowpea is one of the grain legumes that is playing an important role in the livelihood of millions of people in West Africa, especially in Benin. Unfortunately, cowpea on-farm yields are very low. In order to understand the main factors explaining cowpea yield gaps, we collected and analyzed detailed survey data from 298 cowpea fields in Benin during the 2017, 2018 and 2019's rainy seasons, respectively. Composite soil samples were collected from cowpea fields and analyzed in the laboratory. Data on farm field management practices and field conditions were recorded through interviews with 606 farmers. Average cowpea grain yields were low and seldom surpassed 700 kg ha$^{-1}$ on farmer's fields. Significant differences were observed between cowpea grain yields from northern to southern Benin ($p < 0.05$), and the lowest yields were observed in northern Benin. These low yields are related to crop management practices, soil nutrient contents, and the interaction of both. According to the model of regression tree from northern to southern Benin, the use of mineral fertilizer, insecticide sprays to control pests, and the improvement of phosphorus, nitrogen, potassium (P, N, K) and cation sum content in the topsoil would increase cowpea grain yields. Insect pests, diseases, and soil fertility decline are the largest constraints limiting grain yield in Benin. Future research should focus on formulating site-specific fertilizer recommendations for effective cowpea cultivation in Benin, as well as the control of insect pests and diseases.

**Keywords:** food security; soil fertility; crop management practices; cowpea yield





## 1. Introduction

Food security remains a challenge in West Africa, despite numerous opportunities offered by diverse available natural resources [1]. The challenge to food security in West Africa is its underdeveloped agricultural sector that is characterized by low fertility soils and minimal use of external farm inputs, which induce degradation of the environment and low crop productivities [2]. Africa continues to spend about USD 30 to 50 billion per year and will spend about USD 150 billion by 2023 to import food because of insufficient local production [3,4]. Feeding the world in 2050 will not be easy and we need to find ways to boost yields of the main local crops despite the constraints of land, water, fertilizers, and pests.

Among the main local crops, cowpea (*Vigna unguiculata* L. *Walp*.) is one of the grain legumes that is playing an important role in the livelihood of millions of people in West

Africa [5]. Cowpea leaves and grain are substantive foods with protein contents of 27–43% in leaves and 21–33% in grain [6]. Cowpea leaves are also used as livestock fodder in West Africa [7] and they contribute to soil fertility improvement through nitrogen fixation and ground cover [6]. Hence, cowpea plays a major role in human nutrition as a source of protein and is mostly enjoyed with cereal staples, even though it can also be consumed alone. Cowpea seeds are also a major source of vitamins for humans, feed for animals, and a source of cash income [8]. Cowpea has the ability to fix atmospheric nitrogen and grows well on low-fertility soils [9,10]. Most of the world's cowpea (>90%) is grown in sub-Saharan Africa, most of which (83%) is in West Africa [11,12]. In West Africa, Benin, Burkina Faso, Cameroon, Chad, Ghana, Mali, Nigeria, Niger Republic, Senegal, and Togo are the important cowpea growing countries [11,13]. Thus, cowpea is a main food crop and its cultivation remains an opportunity for West Africa to combat food insecurity. Unfortunately, in West Africa, cowpea cultivation, mainly under traditional systems, records low grain yields that seldom surpass 0.3 t ha$^{-1}$ in farmers' fields [11]. Several studies show low cowpea cropping system productivity in West Africa depending on management practices, pest attacks, diseases, low soil fertility and lack of inputs [11,14].

In Benin, cowpea yield on farmers' fields is very low and the national production falls short of meeting the local demand for cowpea seed [15,16]. Cowpea grain yield is low and seldom exceeds 0.5 t ha$^{-1}$ in Benin [17] while the potential grain yields from improved varieties are higher and can reach up to 2.0–2.5 t ha$^{-1}$ and local varieties 0.7–1 t ha$^{-1}$ [18]. This yield gap leads to the following questions: (1) what are the factors that drive cowpea on-farm yields; (2) how can farmers improve cowpea grain yields? We hypothesized that differences between soil fertility management practices, insect pests and disease control, as well as soil physico-chemical characteristics across agro-ecological zones significantly increase cowpea yield gaps in Benin. Our objective is to understand the main causes of variability in yield and to identify opportunities for improving cowpea yield.

## 2. Materials and Methods

### 2.1. Case Study of Villages

The selection of case study villages was based on a rapid regional assessment of the various agro-ecosystems from south to north in Benin among the research and development villages of the National Institute of Agricultural Research of Benin (INRAB). Eleven case study villages that experienced very different agro-ecological (Table 1) and socio-economic conditions were selected: Kokey and Angaradébou (northern Benin); Gbanlin, Yagbo, Miniffi and Kougbadji (central Benin); Adingnigon, Zouzouvou, Adakplamè, Golouhoué and Toulamè (southern Benin). Cowpea cropping systems differed greatly between villages.

In northern Benin, (Kokey and Angaradébou) food production involved cereals (maize and sorghum). Cash crops mainly included cotton, soya and groundnut. Cowpea was sown only once between July and August by most farmers. Based on data from a random sample of 178 farmers, cowpea accounted for 8% of the total area farmed during the 2016–2017 agricultural season [16]. In addition, a large proportion of farmers did not use mineral fertilizers, grew cowpea in pure culture, practiced harness tillage, used crop residues as animal feed, and practiced crop rotation.

In central Benin (Gbanlin, Yagbo, Miniffi and Kougbadji) and southern Benin (Adingnigon, Zouzouvou, Adakplamè, Golouhoué, and Toulamè), food production mainly involved cereals (maize rice and sorghum), tubers and root (yam and cassava), and cash crops (cotton, soya, and groundnut). Cowpea was sown twice and sowing was done in early April, during the long rainy season, and mid-August to early September, during the short rainy season. From a random sample of 204 and 180 farmers in central and southern Benin, respectively, cowpea accounted for 16 and 29% of the total area farmed during the 2016–2017 agricultural season, respectively [16].

**Table 1.** Differences in soil and climate between agro-ecological zones [19–21].

| Characteristics | Southern Benin | Central Benin | Northern Benin |
|---|---|---|---|
| Dominant soil (USDA system) | Ferralsols, and Vertisols | Ferric and Plintic Luvisol | Ferric and Plintic Luvisol |
| Climate type | Subequatorial | Sudano-guinean | North-Sudanian |
| Annual rainfall | 900–1400 mm | 800–1100 mm | 700–900 mm |
| Rainy season | April–mid-July | June–September | June–September |

*2.2. Field Survey*

The total number of cowpea farmers in each village was determined with the help of village authorities using social mapping [22]: 66 and 43 cowpea farmers in Kokey and Angaradébou, respectively, in northern Benin; 59, 63, 60, and 38 cowpea farmers in Gbanlin, Yagbo, Miniffi and Kougbadji, respectively, in central Benin; 42, 52, 78, 67, and 38 cowpea farmers in Adingnigon, Zouzouvou, Adakplamè, Golouhoué and Toulamè, respectively, in southern Benin. All cowpea farmers were surveyed during the 2017 rainy season to identify, in a general way, the different socioeconomic characteristic of the cowpea farmers, and evaluate their experience with cowpea cultivation.

All cowpea fields were surveyed in a random sample of 45, 49, and 41 farmers in southern Benin and 36, 40, and 32 farmers in central Benin during the 2017, 2018 and 2019 rainy seasons, respectively. In northern Benin, all cowpea fields were surveyed in a random sample of 29 and 26 farmers during the 2018 and 2019 rainy seasons, respectively. At the beginning of the growing season, semi-structured interviews were conducted with cowpea farmers whose fields were selected to identify the constraints of cowpea production, the factors driving each farming operation, and the preceding crop (Table 2). During the growing season, semi-structured interviews with these cowpea farmers were conducted on a monthly basis to monitor their management practices, and interview data were cross-validated by our own on-field observations. On each field, five randomly selected $1 \times 1$ m$^2$ plots were staked after sowing to determine the sowing density and estimate cowpea yield. Composite soil samples were taken at the flowering stage. Plots were harvested and cowpea total aboveground biomass was weighed using a hand-held scale with 0.01 g of readability.

**Table 2.** Cowpea cultivation practices from north to south of Benin. Means $\pm$ standard deviations are displayed for continuous variables while proportions are displayed for categorical variables.

| Variables | North | Center | South |
|---|---|---|---|
| Field size (ha) | $0.68 \pm 0.35$ | $1.06 \pm 0.44$ | $0.89 \pm 0.57$ |
| Preceding crop (%) | | | |
| Maize | 32 | 43 | 67 |
| Cotton | 23 | 32 | 13 |
| Sorghum | 21 | 16 | 4 |
| Soybean | 14 | 0 | 0 |
| Millet | 5 | 0 | 0 |
| Fallow | 2 | 4 | 3 |
| Other (Yam, Cassava, rice) | 3 | 5 | 13 |
| Residue management (%) | | | |
| Exported | 89 | 32 | 21 |
| Burned | 2 | 11 | 16 |
| Incorporated | 9 | 57 | 63 |
| Herbicide application prior to land preparation (%) | | | |
| Yes | 34 | 68 | 35 |
| No | 66 | 32 | 65 |

**Table 2.** *Cont.*

| Variables | North | Center | South |
|---|---|---|---|
| Land preparation method (%) | | | |
| Tillage at flat | 86 | 65 | 32 |
| Ridging | 0 | 14 | 57 |
| No tillage | 14 | 21 | 11 |
| Frequency of weeding (%) | | | |
| No weeding | 13 | 19 | 8 |
| Hoe-weeding once | 42 | 54 | 15 |
| Herbicide once | 22 | 0 | 0 |
| Hoe-weeding twice | 19 | 17 | 65 |
| Herbicide once +Hoe-weeding once | 4 | 10 | 0 |
| Hoe-weeding three times | 0 | 0 | 12 |
| Fertilizer application (%) | | | |
| Yes | 16 | 28 | 34 |
| No | 84 | 72 | 66 |
| Applied urea (kg ha$^{-1}$) | $18.5 \pm 5.8$ | $34.54 \pm 10.2$ | $54.2 \pm 7.6$ |
| Applied PNK (kg ha$^{-1}$) | $35 \pm 9.4$ | $47.6 \pm 15.8$ | $68.4 \pm 12.7$ |
| Insecticide application (%) | | | |
| No insecticide use | 41 | 26 | 34 |
| Insecticide once | 24 | 46 | 41 |
| Insecticide twice | 19 | 16 | 22 |
| Insecticide three times | 16 | 12 | 3 |
| Intercropping (%) | | | |
| Yes | 22 | 43 | 65 |
| No | 78 | 57 | 35 |
| Experience with cowpea cultivation (years) | $13.5 \pm 2.8$ | $12.5 \pm 2.8$ | $11.5 \pm 2.8$ |

*2.3. Soil Sampling and Analysis*

Composite soil samples were collected with a soil auger at the depth of 0–20 cm diagonally across 110, 115, and 73 cowpea fields from north to south of Benin during the 2017, 2018, and 2019 rainy seasons, respectively. The soil samples were air-dried, crushed, sieved through a 2 mm mesh screen, and the physicochemical properties were assessed. The pH (water) was measured using the pH meter at a ratio of 1:2.5 soil/water. The organic matter and total N content were determined using the Walkley–Black and Kjeldahl methods, respectively [23]. The available phosphorus was determined using the Bray-1 method [24]. The exchangeable potassium and exchangeable cations were determined by the ammonium acetate method of Metson at pH 7 and atomic absorption spectrophotometry [25].

*2.4. Calculations and Statistical Analyses*

Cowpea grain weight was calculated at 12% moisture content [26]. R software version 4.0.3 [27] was used for statistical analysis. Differences in the average grain yields between zones (northern, central, and southern Benin) during the 2017, 2018, and 2019 rainy seasons were assessed using Kruskal-Wallis tests followed by Dunn tests with Bonferroni as a *p* value adjustment method with the package rstatix [28] since they did not follow a normal distribution. The packages rpart [29] and partykit [30] were used to carry out the regression tree that identifies the determinants of cowpea yield.

The pairwise comparison matrix [31] was used to summarize the data about cowpea production constraints. This pairwise comparison matrix records the proportion of farmers who rank one constraint to the other. Mathematically, the $ij^{th}$ entry of the pairwise comparison matrix is:

$$M_{ij} = \frac{1}{n} \sum_{l=1}^{n} I_{\{w_i^{(t)} < w_j^{(t)}\}}$$

where $M_{ij}$ is the proportion of farmers who rank the constraint $i$ to the constraint $j$; $n$ the number of farmers; $w_i^{(t)}$ and $w_j^{(t)}$ the rank of constraints $i$ and $j$, respectively, according to the farmer $t$; the function $I_{\{w_i^{(t)} < w_j^{(t)}\}}$ is 1 if $w_i^{(t)} < w_j^{(t)}$ and 0 otherwise.

With the pairwise comparison matrix, crossing above or below 50% of farmers involved in this study is a major event; it indicates which constraint is more important than the other [32]. In fact, by counting how many times this threshold was crossed, the constraints are ranked. Multidimensional preference analysis was performed on constraint rank with the package pmr [31].

## 3. Results

### 3.1. Description of Cowpea Cropping Systems

In northern Benin (Kokey and Angaradébou), cowpea was cultivated once per rainy season and started from July to August with land preparation. In more than half of the farmer's fields, cowpea was grown after cereal (Maize, cotton, sorghum), weeds and/or crop residues had been directly incorporated into the soil during tillage (Table 2). Cowpea was usually sown in rows in a single cropping system. Weed control consisted of hoe-weeding and/or applying herbicide. In slightly more than half of the fields, a single hoe-weeding operation was the most frequent weeding method; herbicides were used in 26% of fields. Very few farmers used fertilizer and only at a lower dose of application ($<50$ kg ha$^{-1}$). Pest control activities were performed with insecticides on slightly more than half of the fields. Harvesting methods were manual for all farmers and most of them exported crop residues as animal feed.

In central and southern Benin, cowpea was cultivated twice per year, firstly from April to July (long rainy season) and secondly from August to October (short rainy season). Its cultivation started with field cleaning, i.e., clearing weeds and residues of the preceding crop. In most fields, residues were incorporated into the soil during tillage operations (Table 2). Subsequently, in central Benin, the land was usually prepared by combining herbicide application and manual tillage, while in southern Benin, in most fields, herbicides were not used for land preparation. In central Benin, in more than half of the fields, after land preparation, cowpea was sown in rows in a single cropping system while an intercropping system was mostly adopted in southern Benin. Fertilizers, comprising urea and/or a compound NPK fertilizer, were applied in a few fields right after hoe-weeding and only once in fields with an intercropping system (Table 2). Insecticides were used at least once in most fields for insect pest control. Harvesting methods were manual for all farmers and, in slightly more than half of the fields, crop residues were incorporated into the soil.

### 3.2. Agricultural Practices Driving Cowpea Yields

Average cowpea grain yields were low and seldom surpassed 700 kg ha$^{-1}$ on farmers' fields (Figure 1). Significant differences were observed between cowpea grain yields from northern to southern Benin ($p < 0.05$), and the lowest yields were observed in northern Benin (Figure 1). In northern Benin, fertilization and insect pest control significantly ($p < 0.05$) determined cowpea grain yields on farmers' fields (Figure 2a). The highest cowpea grain yields were recorded on fields where mineral fertilizers were applied while the lowest average yields were recorded on fields without fertilizer application and insect pest control. However, in central Benin, fertilizers NPK, intercropping practice, and insect pest control were the agricultural practices significantly driving cowpea grain yields on farmers' fields (Figure 2b). The highest average yields were observed on fields with NPK fertilizers application in a single cropping system while lowest yields were observed on fields without NPK fertilizers application or insect pest control. An intercropping system led to the decrease of cowpea grain yields compared with a single cropping system. In southern Benin, fertilization and insect pest control were the factors which significantly ($p < 0.05$) determined cowpea grain yields on farmers' fields (Figure 2c). Urea application

strongly influenced cowpea yields in southern Benin and highest average yields were observed in fields where more than 37.5 urea kg ha$^{-1}$ was applied. Insect pest control improved cowpea grain yield compared to no control (Figure 2c).

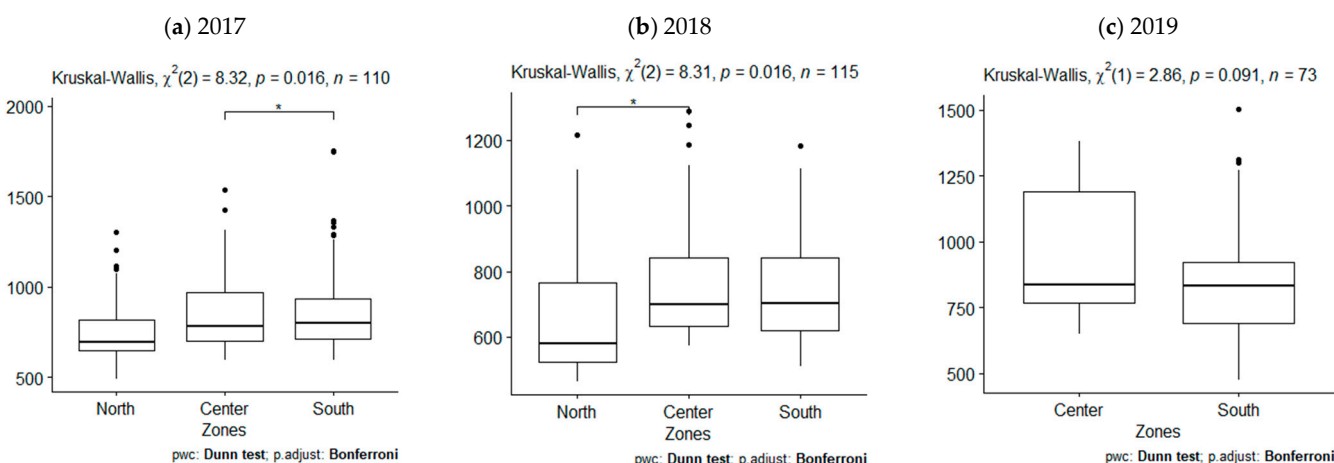

**Figure 1.** Cowpea grain yields on farmer's fields from southern to northern Benin during the 2017 (**a**), 2018 (**b**) and 2019 (**c**) rainy seasons. The y-axes (cowpea grain yield) have different scales. * $p < 0.05$.

### 3.3. Physico-Chemical Characteristics of Soil Driving Cowpea Yields

Soil phosphorus (P) content strongly determines cowpea grain yields on farmers' fields (Figure 3). The highest cowpea grain yields on farmers' fields were significantly ($p < 0.05$) linked to soil P contents and the sum of exchangeable cations in northern Benin, soil P contents and saturation rate in cations in central Benin, and soil P and N contents in southern Benin. In northern Benin, low soil P and N contents significantly ($p < 0.05$) led to the lowest cowpea grain yields, while in central and southern Benin, low soil P, K, and N contents significantly ($p < 0.05$) induced the lowest cowpea grain yield.

### 3.4. Items of Cowpea Cropping System and Environment Responsible for Poor Performance

From northern to southern Benin, farmers identified eight items of cropping system and environment which were responsible for the poor performance of cowpea cropping productivity: insect pests and diseases; soil fertility decline; unavailability of specific fertilizer; unavailability of improved seed; heavy rains; late rain; early drought and weed proliferation. From the pair matrix (Figure 4) in northern Benin, cowpea cultivation constraints were ordered as insect pest and diseases > soil fertility decline > unavailability of specific fertilizer > late rain > weed proliferation > unavailability of improved seed > heavy rains > early drought. In central Benin, constraints were ordered as insect pest and diseases > soil fertility decline > unavailability of specific fertilizer > unavailability of improved seed > heavy rains > late rain > early drought > weed proliferation, while in southern Benin there were insect pest and diseases > late rain > soil fertility decline > unavailability of specific fertilizer > unavailability of improved seed > heavy rains > early drought > weed proliferation. From multidimensional preference analysis (Figure 5), in northern and central Benin the most important constraints were insect pests and diseases, soil fertility decline, unavailability of specific fertilizer and early drought, while in southern Benin, there were insect pest and diseases, soil fertility decline, unavailability of specific fertilizer, heavy rain, and early drought.

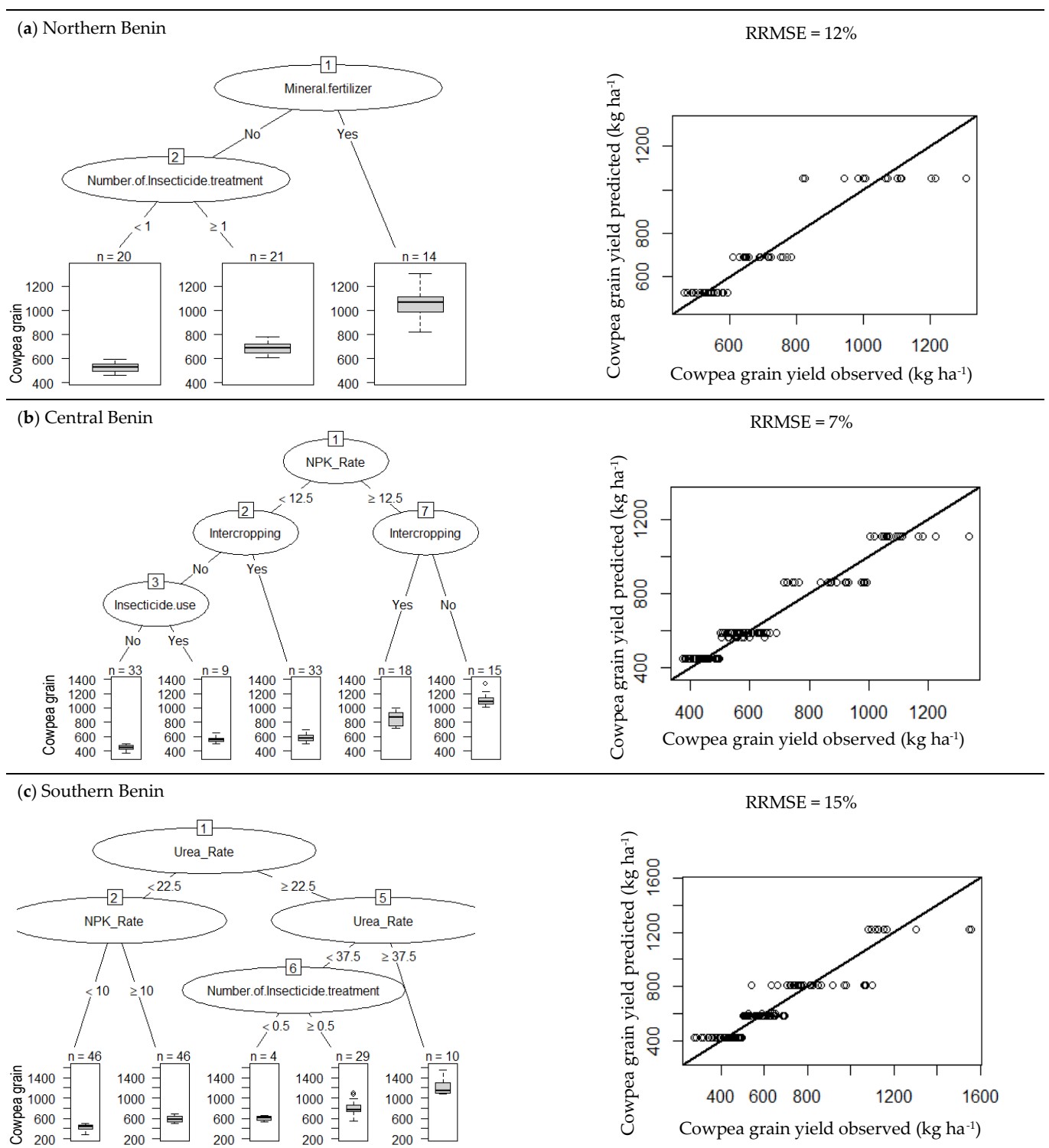

**Figure 2.** Model of regression tree of the agricultural practices driving cowpea grain yields from northern to southern Benin:
(**a**) Northern Benin, (**b**) Central Benin, (**c**) Southern Benin. Urea rate (kg ha$^{-1}$), NPK rate (kg ha$^{-1}$)

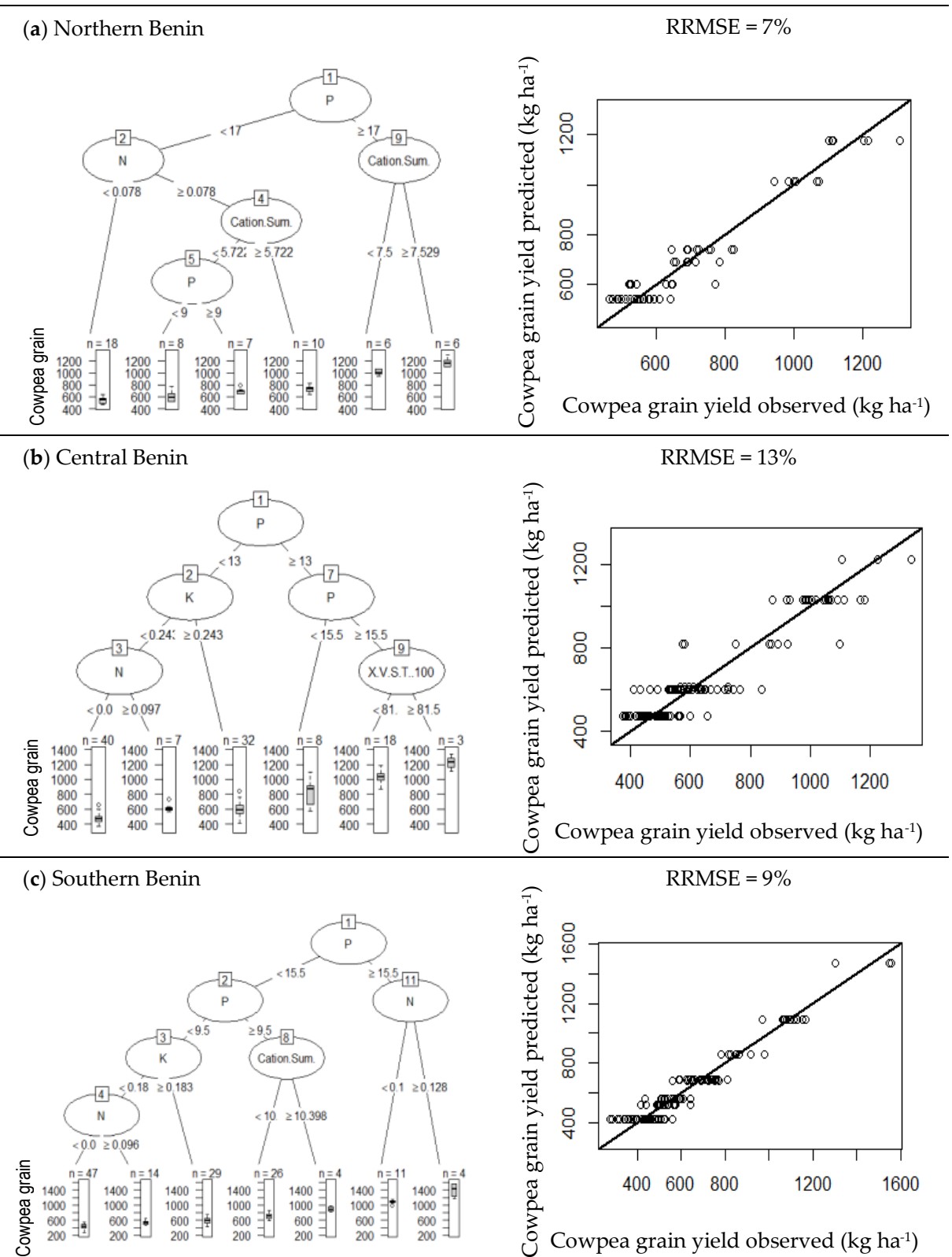

**Figure 3.** Model of regression tree of the agricultural practices driving cowpea grain yields from northern to southern Benin: (**a**) Northern Benin, (**b**) Central Benin, (**c**) Southern Benin. X.V.S.T is base saturation, N (x10 g kg$^{-1}$), P (mg kg$^{-1}$), K (cmol kg$^{-1}$), Cation sum (cmol kg$^{-1}$), X.V.S.T ×100 (%).

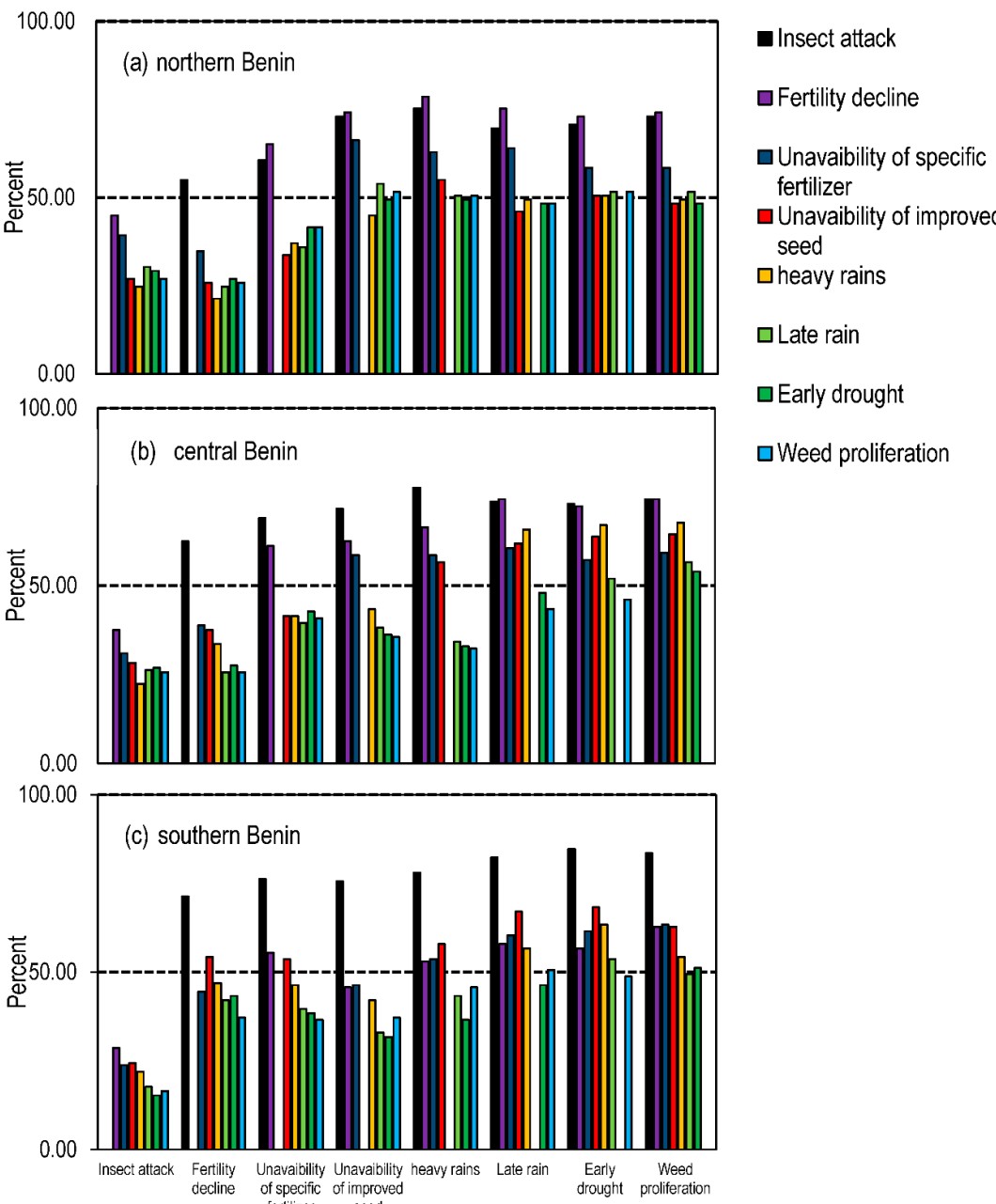

**Figure 4.** Pairwise comparison matrix of items of cowpea cropping system and environment responsible for poor performance from northern to southern Benin: (**a**) Northern Benin, (**b**) Central Benin, (**c**) Southern Benin.

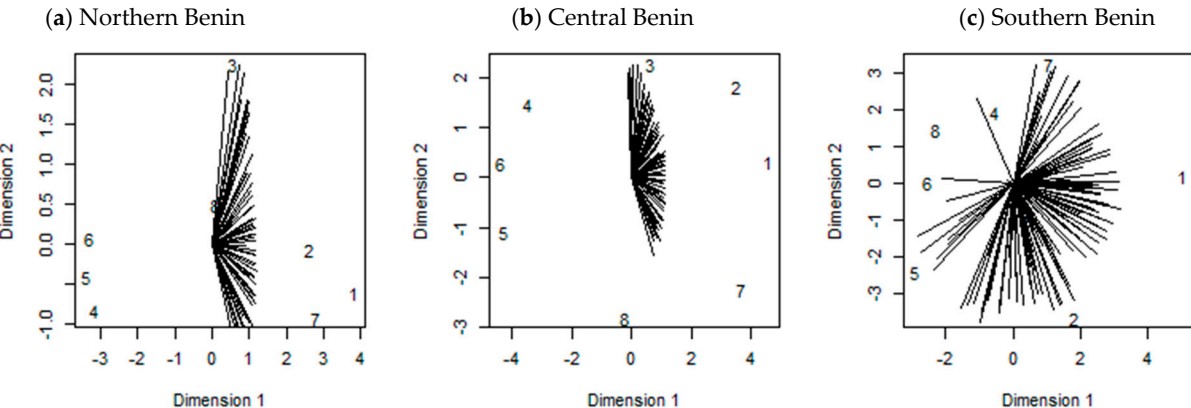

**Figure 5.** Most important items of cowpea cropping system and environment responsible for poor performance from northern to southern Benin: 1 = Insect pest and diseases; 2 = Soil Fertility decline; 3 = Unavailability of specific fertilizer; 4 = Unavailability of improved seed; 5 = Heavy rain; 6 = Late rain; 7 = Early drought; 8 = Weed proliferation. The constraints are labeled with consecutive numbers while the farmers are presented as vectors pointing from the origin to their most important constraint.

## 4. Discussion

In order to understand the main causes of poor performance of cropping systems, variability in yields among cowpea fields, and identify the opportunities to improved cowpea grain yield for enhancing food security, we surveyed cowpea farmers and fields during three crop growing seasons. Our results showed a huge variation in cowpea grain yields, cropping system, factors driving cowpea grain yields and the cowpea cultivation constraints, which suggest the existence of many opportunities to improve cowpea-cropping systems and increased cowpea productivity for food security enhancement in Benin.

### 4.1. Effect of Cropping System on Cowpea Productivity

From northern to southern Benin, when there is no fertilizer, the cowpea grain yields are low. This reveals the low soil fertility impact on cowpea productivity. Unfortunately, in cowpea cropping systems in Benin, fertilizers are rarely used. Our observations from northern to southern Benin indicate that more than 70% of surveyed farmers did not use fertilizer. In addition, where fertilizers were not applied, average cowpea grain yields recorded seldom surpassed 400 kg ha$^{-1}$, while, with the application of approximately 50 and 25 kg ha$^{-1}$ of NPK and urea, respectively, the average cowpea grain yields recorded surpass 800 kg ha$^{-1}$. Insecticide use for pest control and the number of insecticide treatments significantly influenced cowpea grain yield. Our observations suggest that, without insect pest control by insecticide, average cowpea grain yields recorded are low (<450 kg ha$^{-1}$). Kamara et al. [11] reported that major yield-limiting factors of cowpea in West Africa are insect pests, diseases and parasitic weeds. Insect pest attack and disease are major constraints to cowpea production in West Africa [18]. However, our observations indicate that in northern Benin almost half of farmers did not use insecticide even though insect pests attacked cowpea. In addition, and in central and southern Benin, a large part of farmers did not use or used insecticide once. Such insect pest control is not enough to increase cowpea productivity. Cowpea growers in West Africa are at risk of losing the entire crop to insect pests in most growing seasons [11]. Singh et al. [13] reported that insect pests cause the most damage during flowering and podding stages. Therefore, at least two insecticide applications are necessary to reduce the insect population and increase cowpea grain yield. Our results support findings by Singh et al. [13] who noticed that two sprays of insecticide increased cowpea grain yield by 228 and 206% for improved and local varieties, respectively, compared to no sprays of insecticide. Kamara et al. [33] reported that a single insecticide application at the flowering stage increased grain yield by 75%; two applications, once each at flowering and thereafter at pod formation stages,

increased grain yield by 126% in Nigeria. Likewise, Ousmane et al. [18] reported that improved cowpea variety requires 2 to 3 insecticide sprays to control major pests. Due to the photosensitivity of cowpea varieties grown by the farmers, the sowing is late to improve early flowering [11]. The late planting, especially in central and southern Benin, allows cowpea plants to flower when various insect pest population rise. In central and southern Benin, the majority of farmers grow cowpea in intercrop with other crops such as maize, pearl millet, sorghum and cassava. Our study shows that this intercropping of cowpea with these crops reduces cowpea grain yield compared to the sole cropping of cowpea. Even though cowpea is grown as an intercrop with these crops, farmers do not use or use low fertilizer. Therefore, cowpea plants and these cereals compete for the low nutrient content in the soil. In addition, in intercropping systems, cowpea sown density is lower than that of sole cropping of cowpea. This low density could also explain the low yield obtained in the intercropping of cowpea [34]. Our results support findings by Olufajo and Singh [35] who reported that the intercropping of cowpea with maize has a major weakness of very low cowpea yield. According to these authors, the major reason for the low cowpea yields in intercropping systems is shading by taller cereal plants.

*4.2. Strategies to Improve Cowpea Productivity*

The average cowpea yields in northern, central, and southern Benin were $525.2 \pm 211.4$, $824.9 \pm 221.7$, and $758.8 \pm 235.6$ kg ha$^{-1}$, respectively. The results of this study show that those great differences relate to crop management practices, soil nutrient contents and the interaction of both from northern, central and southern Benin, respectively. According to the model of regression tree from northern to southern Benin, cowpea grain yield would be improved by crop management practices such as the use of mineral fertilizer and insecticide sprays to control pests. In addition, increasing P, N, and cation sum content into the topsoil would improve cowpea grain yield. Furthermore, in central and southern Benin, increasing K content into the topsoil would improve cowpea grain yield. However, intercropping systems would reduce cowpea grain yield in central Benin. These results show that crop management practices, which improve N, P, K, and cation sum content into topsoil, on the one hand and on the other, reduce major insect pest damages, and significantly increase cowpea grain yield. Our results support findings by Bationo et al. [36] who reported that the cowpea yields are very low due to several constraints including poor soil, insect pests, and drought. In Benin, cowpea farmers do not use fertilizer in the sole cropping of cowpea [16] because it fixes its own nitrogen from the air using the nodules in its roots [8]. Our study suggests that N, P, and K are needed to apply as fertilizer for increasing cowpea yield. In fact, in areas where soils are poor in nitrogen, there is a need to apply a small quantity of nitrogen as a starter dose [8]. Such results are noticed by Bationo et al. [36] who reported significant cowpea responses to nitrogen applied as urea in different agro-ecological zones of the West African savannahs [36]. However, cowpea does not fix enough nitrogen because of limitations in other crop nutrients. Among those, P is critical to cowpea yield because it stimulates growth, initiates nodule formation, promotes rhizobium-legume symbiosis, improves pod formation as well as cowpea yield [37–39]. Phosphorus deficiency is the most limiting soil fertility factor for cowpea production in many tropical soils where the total available P levels are very low [8,11]. Consequently, cowpea cultivation without adequate fertilization and insecticide sprays for insect pest control does not offer any hope of higher yields. Unfortunately, no fertilizer formulation is recommended for cowpea production in Benin and farmers continue to grow cowpea mainly without fertilization [16]. Our study shows that cowpea farmers are aware of these major constraints, which ranked highest from the north to the south of Benin.

**5. Conclusions**

The common analysis of the relative gap between field and potential yields of cowpea was expanded with an analysis of the factors driving cowpea yields in farmer fields and the major constraints of efficiency of cowpea cultivation. The approach was based on the

assumptions that improvements in farmer cropping practices and soil fertility constitute the keys to increasing cowpea yields and that an increase in cowpea productivity is a key to stimulating the expansion of cowpea growth for assuring food security.

From northern to southern Benin, cowpea grain yields are very low and depended on management practices such as the use of mineral fertilization, insecticide sprays to control pests, and phosphorus, nitrogen, potassium and cation sum content into the topsoil. Insect pests, diseases and soil fertility decline are the major constraints that limit cowpea cultivation in Benin. Future research should focus on formulating fertilizer requirements, and insect pests and disease control for effective cowpea cultivation in Benin.

**Author Contributions:** This work was carried out in collaboration among all authors. Author F.N.A. designed the study, wrote the protocol, collect the data and soil samples, and wrote the first draft of the manuscript. Author E.C.A. and B.T.C.O. managed the analysis of soil samples at the laboratory and performed the statistical analysis. Authors G.D.D. and L.G.A. managed the literature searches. All authors have read and agreed to the published version of the manuscript.

**Funding:** This study is financially supported by Smallholder Agricultural Production Enhancement Program (SAPEP).

**Institutional Review Board Statement:** Not applicable.

**Informed Consent Statement:** Not applicable.

**Data Availability Statement:** The data presented in this study are available on request from the corresponding author.

**Conflicts of Interest:** The authors declare that they have no competing interests.

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
