# Peer review of "Cultivation of Cowpea Challenges in West Africa for Food Security: Analysis of Factors Driving Yield Gap in Benin"

_agronomy, doi:10.3390/agronomy11061139_

Round 1

Reviewer 1 Report

This paper reports on the results of a survey conducted on the causes of low yield of cowpea across three agroecological zones in Benin. Insect pest, diseases and the declining soil fertility were the main causes identified by exploratory analysis. Provide soil classification and climatic regimes by zone. I confess that I am not familiar with the statistical methods used in this paper. With this kind of study, I would use machine learning methods.

After presenting the big picture in figure 1, the three agroecological zones could have been analyzed separately to focus on and discuss problems specific to each zone. Paragraphs are often too lengthy, driving the reader away from the mainstream idea.

Introduction: provide some literature to document features listed in Table 1, especially those tested in Figures 2 and 3. What are the main features likely affecting cowpea yield that should be addressed in Benin? L. 222-227, l. 268-270 and l. 280-284 should be moved to introduction to support hypotheses.

l. 54: replace ‘poor soils’ by ‘low-fertility soils’

l. 68-72: formulate questions in terms of hypotheses that can be accepted/rejected by statistical tests. Write your objective as ‘Our objective is to …’.

l. 75-77: provide a Table showing differences in soil and climate between the three agroecological zones.

There were 178  farms in northern Benin, 204 farms in central Benin, and 180 farms in southern Benin (l. 86 and 95), totalling 562 cases. However, those figures appeared not to be compatible with figures reported l. 101, 103, 104, 108, 111, and 123. Please clarify.

l. 112 and 115: privode a list of questions asked to farmers in the so-called ‘semi-structured’ interviews as supplement materials. There seems to be 30 variables, mostly managerial, in Table 1. There is no climatic variable.

l. 117-119: not clear (the sentence should be reversed because the flowering stage occurs after sowing.

l. 120: provide the precision of the hand-held scale.

l. 124: sieved to what particle size?

l. 126-130: provide references for methods. How was Bray1-P quantified.

l. 135-136: why did you use the Kruskal-Wallis non-parametric test followed by the Dunn-Bonferroni parametric test?

l. 138: how did you conduct your regression tree analysis? Did you use machine learning methods? What features did you include in your model?

l. 138: why to identify determinants of rainfed rice while cowpea only was surveyed?

l. 140-151: provide a reference for ‘pair matrix’. Do you mean correlation matrix? It is not clear how to run such computation from survey data. How do you rank constraint i to constraint j, what are the ranks of constraints, what are the constraints? What is the meaning of conducting multidimensional reference analysis on constraint ranks?

l. 154-156: cowpea grown during the rainy season and maize during the dry season? Maize is very demanding in water and nutrients.

l. 200: how did you obtain cutoff urea rate of 37.5 kg urea/ha?Converted to N, it means 17 kg N/ha?

Figure 2a: highest yield obained with minerla fertilization but without insecticide application?

Figure 3: how did you opt for the cutoff values for soil test values shown in the figure? What is XVST? What cations were added to cation sum? Organic matter was quantified (l. 127) but does appear in the figure. Is there a reason?

Figure 4 is unreadable.

Figure 5: not clear how heavy rain and drought were included into the model. Not clear how to read that figure.

l. 302-311: do you mean that intercropping should be adandoned? Why is intercropping promoted in Africa?

l. 358-359: do you mean soil test calibration against crop response (expensive but informative), or simply fertilizer trials to derive response curves averaged across experimental sites (low-cost, much less informative)?

Reviewer 2 Report

-The paper looks good, but really needs some grammar checks, many places are missing commas, some sentences are incomplete, etc. It needs to be fixed. The citations need to checked to make sure they in order and acceptable by the journal, and be consistent by writing "central Benin" in replace of "Center Benin" in the texts and even on the figures. I am attaching the manuscript with my comments on it. Please review and address.

Round 2

Reviewer 1 Report

Several editorial corrections are required as follows:

  1. 15: to find the…
  2. 21: in the laboratory
  3. 29: and the largest constraints limiting grain yield
  4. 30: formulating site-specific fertilizer recommendations…
  5. 63: reveal?
  6. 65: lack of inputs
  7. 69: seldom exceeds?
  8. 71: reformulate correctly
  9. 82, 85: reformulate?
  10. 102: central
  11. 111: in each village
  12. 114, 120: central Benin
  13. 113, 114, 116, 122, 138: , respectively, (commas)
  14. 124: delete ‘to field survey’
  15. 125: delete ‘identify’
  16. 136: a soil auger
  17. 142-143: delete ‘according’ and replace by ‘using the’
  18. 148: statistical analyses (delete ‘data’)
  19. 164: define ‘significant’, at what level of significance?
  20. 177: lower than which dose?
  21. 178: of the fields
  22. 179: as animal feed

l, 182: secondarily

  1. 187: for land preparation
  2. 188: the fields
  3. 205, 231: northern (be consistent)
  4. 206: significant at what level of significance?
  5. 211: agricultural
  6. 215: compared with
  7. 216: that significantly (p = 0.05 ?)
  8. 218: in fields
  9. 227-228: reformulate
  10. 229, 232, 233: indicate probability after significantly
  11. 327: systems
  12. 329: delete ‘and studied’
  13. 331: rephrase as ‘variation in cowpea grain yields’…
  14. 340: reveals
  15. 341: Unfortunately
  16. 343: surveyed farmers
  17. 343: where not ‘when’
  18. 344-345: surpassed
  19. 354: insecticide once (inversion)
  20. 357: plodding?
  21. 358, 373: then?
  22. 359, 377, 395: Our results support findings by …
  23. 362: a single insecticide application…
  24. 363: and thereafter at pod (?) stage
  25. 374: rival or compete?
  26. 380: shading by
  27. 382-383: present yield means and standard deviations as well as least significant difference
  28. 383: ha exponent (-1)
  29. 384: differences related to
  30. 404: put period before ‘among’
  31. 405-406: stimulates, initiattes, promotes, and improves
  32. 411: for insect pest
  33. 412: no ferrtilizer formulation
  34. 413: continue to
  35. 415: ranked
  36. 419: expanded
  37. 422: increase
  38. 424: southern
  39. 425: mineral fertilization
  40. 426: phosphorous or phosphorus? Be consistent.
  41. 427: highest major? Or simply: major?

Author Response

Response to Reviewer comments

We are grateful to the editors and the reviewers for their suggestions. We have revised the manuscript to take into account the suggestions and provided a response to the comment.

Point 1. Several editorial corrections are required as follows: …

Response 1

We proofread the manuscript and corrected point-by-point, the several editorial corrections as requested.